# The Influence of Separate and Combined Exercise and Foreign Language Acquisition on Learning and Cognition

**DOI:** 10.3390/brainsci14060572

**Published:** 2024-06-03

**Authors:** Yijun Qian, Anna Schwartz, Ara Jung, Yichi Zhang, Uri Seitz, Gabrielle Wilds, Miso Kim, Arthur F. Kramer, Leanne Chukoskie

**Affiliations:** 1Department of Physical Therapy, Movement and Rehabilitation Science, Bouvé College of Health Science, Northeastern University, 360 Huntington Ave, Boston, MA 02115, USA; qian.yiju@northeastern.edu (Y.Q.); schwartz.ann@northeastern.edu (A.S.); jung.ara@northeastern.edu (A.J.); g.wilds@northeastern.edu (G.W.); 2College of Art, Media and Design, Northeastern University, 360 Huntington Ave, Boston, MA 02115, USA; zhang.yichi6@northeastern.edu (Y.Z.); seitz.u@northeastern.edu (U.S.); m.kim@northeastern.edu (M.K.); 3The Center for Cognitive and Brain Health, Northeastern University, 805 Columbus Ave, Boston, MA 02115, USA; a.kramer@northeastern.edu; 4Beckman Institute, University of Illinois, Urbana, IL 61801, USA

**Keywords:** learning, cognition, exercise, language, dual-tasking, older adults

## Abstract

Aging contributes significantly to cognitive decline. Aerobic exercise (AE) has been shown to induce substantial neuroplasticity changes, enhancing cognitive and brain health. Likewise, recent research underscores the cognitive benefits of foreign language learning (FLL), indicating improvements in brain structure and function across age groups. However, the lack of a comprehensive paradigm integrating language learning with exercise limits research on combined effects in older adults. In order to address this gap, we devised a novel approach using a virtual world tourism scenario for auditory-based language learning combined with aerobic cycling. Our study examines the impact of simultaneous AE and FLL integration on cognitive and language learning outcomes compared to FLL alone. A total of 20 older adults were randomly assigned to AE + FLL and FLL-only groups. The results revealed significantly improved Spanish language learning outcomes in both combined and language learning-only groups. Additionally, significant cognitive function improvement was observed in the FLL group following short-term language learning.

## 1. Introduction

The efficacy of aerobic exercise (AE) and cognitive training (CT) programs to enhance cognitive health among older adults has received considerable attention. Extensive research has consistently shown that AE leads to significant structural and functional neuroplasticity, resulting in enhanced cognitive functions [1,2,3] and an improved sense of overall well being [4]. The findings from the effects of CT programs, on the other hand, show mixed results. Cognitive training programs, while effective at improving the specific skills targeted in training, often do not result in meaningful transfer to untrained tasks [5,6]. This discrepancy may stem from the inherent specificity of traditional CT methods, which typically target isolated cognitive skills rather than encompassing a broader range of cognitive functions, such as attention [7] and working memory [8], reflective of daily tasks.

Recent research has highlighted the potential cognitive benefits of foreign language learning (FLL), revealing improvements in brain structure and function across both young and older adults [9,10]. Given these findings, the prospect of FLL serving as a viable strategy for providing resistance to some aspects of cognitive aging [11,12,13] presents itself as an alternative to traditional CT programs. FLL entails a complex learning process that engages a diverse array of cognitive skills [7,8], suggesting that it serves not only as a form of acquisition but also as a mode of challenging cognitive training.

The concurrent or simultaneous performance of physical exercise and cognitively challenging activities is known as combined, multidomain, or dual-task training [14,15,16]. Research on dual-task performance has a long tradition of investigating how increased attentional demands affect either cognitive or physical performance due to prioritization in resource allocation to one or the other domain. Thus, these paradigms assume that our information processing system is limited and that conflicts in resource allocation are solved via interference control [17]. Neuroscientific approaches examine the structural and functional brain resources that support physical and cognitive activities, both separately and together, and the manner in which these resources change with practice and learning [18]. In recent years, research has delved into the synergistic effects of combining exercise and cognitive training, which have demonstrated greater efficacy in promoting cognitive function and enhancing learning [2] than single training alone [14,19,20,21]. However, most AE combined and single training alone studies apply traditional CT, and several studies demonstrate that moderate exercise facilitates learning in younger populations [22]. Still, limited investigations explored the combination of AE and FLL in older adults [23].

Even though the benefits from combined training seem promising, the lack of older adult studies in such a strategy might be due to the concern of an overwhelming combination of tasks that engage physical and mental demands. Previous studies have examined the preferences and challenges faced by older adults during exercise, travel, and studying foreign languages [24,25,26]. Researchers have recommended designing game-related programs for older adults and individuals requiring additional support by simplifying game objectives, providing interactive systems [27] and timely feedback [28] to support learning, and incorporating appropriate visual complexity [29].

Thus, we address this challenge by creating a virtual world tourism scenario gamified with language learning and exercise, combining auditory-based language learning with aerobic cycling [30]. In this study, we aim to explore how aerobic exercise influences foreign language learning and cognitive skills.

## 2. Materials and Methods

### 2.1. Participants

Twenty participants aged from 65 to 85 years old were recruited through posters in the Boston area, senior centers, and bulletin boards in the local community, as well as through email and phone calls. Potential participants were evaluated for eligibility through phone screening or online screening survey questionnaires. All participants were generally healthy, without significant physical impairments or visual or hearing acuity issues. Those with a history of neurological diseases (Alzheimer’s, Parkinson’s, epilepsy, chronic migraines, multiple sclerosis, and stroke) or heart-related conditions (unstable angina, heart attack, heart failure, arrhythmia, and valve diseases) were excluded. People with high blood pressure with medication were included. All participants had minimal Spanish exposure, with no formal Spanish learning experience. Participants who regularly had more than 60 min of moderate exercise, such as cycling and jogging, per week in the past 6 months were excluded. We used standard cut-off scores for older adults on the Montreal Cognitive Assessment (MoCA ≥ 24) and Kaufman Brief Intelligence Test Second Edition (KBIT-2 ≥ 85) to ensure that the participants were qualified for the integral cognitive and intellectual demands of our experiments. This study was conducted in accordance with the ethical principles of the Declaration of Helsinki. Ethical approval was obtained from the Institutional Review Board of Northeastern University, and written informed consent was obtained from all participants prior to their inclusion in the study.

### 2.2. Procedure

The complete experiment involves six visits over approximately 19 days. During the initial 90–120 min visit, eligible participants had their blood pressure measured, received a detailed explanation of the experimental procedures, and signed consent forms. A demographic survey was conducted to collect information on age, gender, education, marital status, and income. Participants also took a pretest of their Spanish knowledge.

After the first visit, the participants were randomly assigned to either the AE + FLL group (biking and language learning) or the FLL group (language learning only). For the second to fifth visits, the participants engaged in auditory-based immersive Spanish learning with or without stationary biking. The participants in the FLL group were also asked to sit on the stationary bike to experience the exact auditory-based learning material (virtual city route) as the AE + FLL group but were asked not to pedal during the entire training, see Figure 1.

Researchers provided Polar H10 and Scosche Rhythm 24 heart rate monitors, adjusted bike seats, and calculated the target heart rate range based on age (64% to 76% of (220-age)). The lab streaming layer (LSL) was used to collect Polar H10 raw ECG data for future analyses. The Elite HRV was operated on an LG phone and was placed within the participants’ line of sight so they could monitor their heart rates in real time. In the AE + FLL group, the participants pedaled while maintaining their heart rate within the target range for 30 min. A researcher supervised each session, handling the equipment, monitoring for emergencies, and providing feedback during mini quizzes. Each of the second to fifth visits lasted 60–90 min. The NASA-TLX was given to participants immediately after they finished the fourth training session. After the fifth visit, the participants were invited back to the lab for a sixth visit to complete another round of cognitive tests and the Spanish knowledge post-test.

### 2.3. Virtual System Design

The virtual system was developed by modifying the open-world game Grand Theft Auto 5 (GTA 5) using the C# programming language. We selected GTA 5 due to its extensive and detailed geographical content, offering a high-quality representation of landmarks. Third-party mods, including gta5-real-mod, were incorporated to create a realistic 2D virtual city bike tour for participants. The narrative script and Spanish audio were generated using the Voicemaker website. We recorded footage from GTA 5 and then edited this in Adobe Premiere to incorporate AI-generated audio and visual cues of landmark highlights.

### 2.4. Language Learning

The virtual bike tour is in Castilian Spanish and is divided into episodes, blending exercise games to keep participants engaged. The tour offers immersive visuals and auditory cues while monitoring heart rates. The tour scripts include Spanish phrases spoken by a “biking buddy” who prompts participants to repeat after him. Our approach relies on older learners’ ability to absorb language through pattern recognition, even without direct teaching. This learning environment is designed to mimic many features of how children acquire their first language. It contains no written text, relies entirely on auditory processing, and conveys information only by the learner hearing speakers using the same structures repeatedly. Importantly, this method has been demonstrated to be a feasible way for adults to extract and acquire semantic and syntactic patterns from artificial language [32].

The language learning is divided into two parts in each training session, with four training sessions in total. The first part is in the virtual system with 30 min of auditory-based immersive Spanish learning immediately followed by the post-session test, which also serves as additional learning reinforcement. The post-session test learning materials include the phrases the participants heard from the current training session, and to be consistent with the learning mechanism, the question prompts and choices are either audio or pictures related to the Spanish knowledge points that participants heard or observed or the audio they heard from the training sessions. All questions are multiple-choice questions, and the participants are told to choose the best answer to the prompt. Participants will receive the correct answer if they answered wrong and be allowed to review the answer if they answer correctly. The goal is to consolidate the participants’ learning results.

Targeted Spanish knowledge was carefully selected for learners with minimal exposure to Spanish and categorized into the recognition of nouns and daily phrases, as well as several syntactical patterns in Spanish that differ from English: gender agreement with article, reversal of adjectival agreement with noun gender relative to English word order, the use of prepositions, and some basic contrasts involving pronouns/conjugations.

During the virtual immersive learning, participants’ exposure to Spanish was entirely auditory-based with no visual text display. The structure of the Spanish was highly repetitive and contrasting pairs were used to help learners identify patterns. Each phrase was presented to participants in brief chunks of a few words at a time, which they were encouraged to repeat immediately, and they would then hear the full sentence again. After they had heard each target phrase with a particular noun or noun phrase (e.g., “la iglesia”) twice, they would be prompted with a one-question comprehension quiz to receive some feedback on the patterns they were extracting, see Figure 2.

The program consists of four, 30-min cumulative virtual bike routes. Three nonplaying characters (NPCs) accompany the participant on the tour, see Figure 2.

Joshua (Narrator): A male native English speaker who acts as the tour guide in the game. He introduces landmarks to participants and gives detailed descriptions of landmarks in English, but with no direct correspondence to the Spanish, except in the name of the landmark.Lorenzo (“Bike Buddy”): A male native Castilian Spanish speaker who acts as the local Spanish tour guide and a Spanish instructor in the game. He speaks Spanish phrases to participants according to which landmarks are coming up and directs participants to repeat after him to learn.Ashley (Companion): A female native English speaker who acts as the travel companion who is trying to learn Spanish in the game. She frequently requests that Lorenzo repeat Spanish sentences for participants. She also helps participants to engage actively in the learning process by restating parts of Lorenzo’s phrases to “make sure she is understanding”. These statements provide the context of the in-game mini quiz for participants, and they involve examples of English-accented Spanish.

In our approach, the participants extract the meaning of the Spanish phrases by listening to cues from the English narration, observing the environment, such as highlighting landmarks, and noting what parts of the Spanish phrases are consistently repeated and which are not relative to events on the tour. Each 30 min virtual bike route includes 10 landmarks, such as churches, banks, or golf courses. At each landmark, a highlight appears to guide participants’ attention to that landmark. The goal is to help learners recognize patterns in Spanish syntax as they listen and engage with the virtual tour.

### 2.5. Measurements

#### 2.5.1. Learning Performance

The participants’ accuracy rates on the Spanish pre- and post-tests were recorded. These tests encompassed various vocabulary terms and probed each syntactical function that served as the learning objective of a lesson during the experiment. Both the pre- and post-tests followed a similar format, with adjusted question prompts to minimize exposure bias. They evaluated the same language knowledge, covering questions on nouns, noun genders, adjective order, prepositions, and pronouns. The accuracy rate from the pretest served as a baseline for comparison with the participants’ post-test accuracy rate. The post-test directly assessed the language, including vocabulary targets and syntactical rules, to which the participants were exposed.

#### 2.5.2. Cognitive Performance

A battery of cognitive tests was administered, including the Montreal Cognitive Assessment (MoCA), the Kaufman Brief Intelligence Test Second Edition (KBIT-2), and cognitive tests from the NIH Toolbox [33]. These included the Flanker Inhibitory Control and Attention Test (Attention), the List Sorting Working Memory Test (WM), and the Pattern Comparison Processing Speed Test (PS). The scores from the cognitive battery tests were age-corrected and based on norm-referenced, standardized assessments commonly used in both research and clinical settings. The pretraining cognitive function tests (Attention, WM, and PS) served as a baseline for comparing post-training cognitive performance to assess potential improvement.

#### 2.5.3. Workload Measurement

Additionally, the NASA Task Load Index (NASA-TLX), a workload measurement scale was utilized to evaluate the cognitive loads induced by the task [34]. This scale comprises six dimensions: Physical Demand, Mental Demand, Performance, Effort, Temporal Demand, and Frustration.

Mental demand (MD): How much thinking, deciding, or calculating was required to perform the task;Physical demand (PD): The amount and intensity of physical activity required to complete the task;Temporal demand (TD): The amount of time pressure involved in completing the task;Effort: How hard does the participant have to work to maintain their level of performance;Performance: The level of success in completing the task;Frustration level: How insecure, discouraged, secure, or content the participant felt during the task.

NASA-TLX is widely utilized to assess users’ experiences in multitasking environments, encompassing both physical and digital tasks. While NASA-TLX involves rating and weighting, weights are not commonly applied due to individual differences in the importance of dimensions. A recent study has indicated that rating scores alone are more meaningful in workload measurement, and in most scenarios, they should be treated as intervals in population analyses and be ordinal for individuals [35]. Therefore, we utilized rating scores to evaluate participants’ level of engagement and the efforts they exerted between groups.

### 2.6. Data Process

The raw pre- and post-Spanish accuracy data were collected using Qualtrics; then, they were organized in Excel (version 16.74) and exported as a CSV file. Data analysis and visualization were performed using R Studio (version 4.3.3). During the analysis, the post-cognitive performance data of one participant in the AE + FLL group was excluded due to an unexpected event: the participant had to answer an emergency call during the post-cognitive function measurements, which compromised the consistency of the measurements. Furthermore, NASA-TLX measurements were collected starting from the sixth participant, and one participant’s NASA-TLX score was excluded because of the application error. Therefore, the NASA-TLX analysis only includes data from 14 participants.

### 2.7. Statistical Analysis

The normality of the data was assessed using the Shapiro-Wilk test. The results indicated that both the pre- and post-Spanish accuracy, as well as accuracy improvements, were normally distributed in both groups. However, in the AE + FLL group, post-intervention attention test scores were found to deviate from a distribution. Similarly, in the FLL group, the preprocessing speed and changes in attention score did not adhere to a normal distribution. Learning improvements were computed by subtracting the baseline score measured before interventions from the post-accuracy condition, and the same procedure was applied for cognitive improvements.

For the AE + FLL group, a one-tailed paired sample *t*-test was utilized to evaluate increases between participants’ pre- and post-Spanish test accuracy. The same test was also applied to assess processing speed for the AE + FLL group, while a paired Wilcoxon signed-rank test was used to analyze improvements in attention. In the FLL group, a one-tailed paired sample *t*-test was used to examine learning performance and attention and working memory performance before and after interventions. A Wilcoxon signed-rank test was used to analyze any improvements in processing speed in the FLL group.

In order to evaluate the impact of aerobic exercise on language learning and cognitive performance between groups, various statistical tests were conducted. A two-tailed independent *t*-test was employed to assess differences in Spanish accuracy improvements, processing speed, and working memory improvements between the AE + FLL group and the FLL group. Additionally, a Wilcoxon rank-sum test (Mann–Whitney U test) was used to measure differences in attention improvements between the groups.

The NASA-TLX ratings, including Mental Demand (MD), Physical Demand (PD), Temporal Demand (TD), Performance, Effort, and Frustration, were compared between groups. The normality of the data was assessed using the Shapiro-Wilk test. An unpaired *t*-test was conducted to assess the difference between MD, TD, Performance, Effort, and Frustration between groups, while the Wilcoxon rank-sum test was used to assess PD.

## 3. Results

### 3.1. Participants

The participants were randomly assigned to the biking language learning group (AE + FLL Group, n = 10) or the language learning-only group (FLL Group, n = 10), resulting in a total of 20 participants (nine female and 11 male) aged 65 to 85 (M = 73.8; SD = 4.1). We observed no significant differences between the two groups in terms of age, gender, global cognitive function (MoCA) scores, IQ (KBIT) scores, existing Spanish proficiency, cognitive performance before intervention, average training session completion time, and the delay (in days) between the last training session and post-measurement (Table 1).

### 3.2. Learning Performance

#### 3.2.1. Within Group Comparison

The Spanish learning outcomes significantly improved in both the AE + FLL group (*t* = 7.18, df = 9, *p* < 0.0001 ****) and the FLL group (*t* = 8.16, df = 9, *p* < 0.0001 ****) after four sessions of training; see Figure 3.

#### 3.2.2. Between Group Comparison

No significant differences were found between the AE + FLL group and the FLL group in terms of pre-Spanish (*p* = 0.4) and post-Spanish knowledge (*p* = 0.48). Additionally, we did not find significant differences between the groups in terms of the improvements in Spanish knowledge (*p* = 0.69); see Figure 3.

### 3.3. Cognition Performance

#### 3.3.1. Within Group Comparison

Both groups show a trend of increased cognitive performance after four sessions. The FLL group shows statistical differences in the attention test (*t* = −2.45, df = 8, *p*-value = 0.01 **) and PS test (*V* = 6, *p*-value = 0.01 **); see Figure 4. However, only the AE + FLL group shows a nonsignificant trend of improvements in working memory (*t* = −1.69; df = 8; *p*-value = 0.06); see Figure 4.

#### 3.3.2. Between Group Comparison

The changes in cognitive performances, including attention, processing speed, and working memory, were compared between the AE+FLL group and the FLL group, and no significant differences were observed between the groups; see Figure 5.

### 3.4. Workload

Overall, the perceived mental demands (MDs), performance, and effort levels were similar between the AE + FLL group and FLL group. The FLL group exhibited a slightly higher level of frustration compared to the combined group; however, this difference was not statistically significant. Perceived physical demands (PDs) were significantly different between the groups (*W* = 32; *p* = 0.007 **), with the AE + FLL group reporting higher physical demands during task completion. Additionally, the AE + FLL group also perceived higher temporal demands (TDs) (*t* = 2.51; df = 9; *p* = 0.03 *) compared to the FLL group; see Figure 6.

## 4. Discussion

We present the design and preliminary data of a novel paradigm of integrating exercise with an audio-based language learning program. Diverging from traditional text-based approaches, we adopted a ’listen and repeat’ method inspired by children’s language acquisition for our audio-based learning.

In this study, we aimed to investigate how exercise influences foreign language acquisition by directly measuring learning outcomes and cognitive effects. Our findings demonstrate improvements in learning regardless of whether participants engaged in exercise alongside language learning. Across both groups, there is a median trend of improved cognitive skills after four training sessions. Notably, only the group engaged in foreign language learning exhibited more significant changes in attention and processing speed compared to the baseline. In line with prior research suggesting the benefits of aerobic exercise for working memory [36,37,38], our results indicate that the exercise-involved group shows almost significant changes in working memory. However, we did not detect significant differences in cognitive performance changes between groups, suggesting that aerobic exercise during foreign language learning tasks may yield comparable learning and cognitive benefits. It is worth mentioning that the upper and lower limits for attention and PS in the FLL group are higher. When examining these markers across the AE + FLL condition, the upper and lower limits vary. Thus, it is possible that we did not observe significant group improvements in the post-cognitive measurements, but improvements likely occurred based on individual differences, which require further measures.

The question arises as to why simultaneous moderate exercise and language learning yield equivalent learning effects compared to the language learning-only group while showing varying results in promoting cognitive functions, especially the distinction between attention and processing speed benefits from foreign language learning (FLL) and working memory might be benefited from the combined strategy. A proposed hypothesis to explain this phenomenon is that exercise diverts resources necessary to perform cognitive tasks during training, potentially leading to interference. It is worth mentioning that in our paradigm, exercise overlaps with language learning during training, and all participants were in a low-active condition. In this scenario, the mental workload required by exercise may be too high to effectively manage the cognitive task [39]. Cognitive loads were assessed using NASA-TLX, revealing that the participants in the combined group perceived significantly higher physical demands. Dividing attention to maintain a moderate exercise level could potentially limit engagement in cognitive tasks and the ability to identify patterns between virtual environments and Spanish phrases.

It is also possible that the benefits of physical exercise in terms of neuroplasticity may not be immediately evident [40]. However, with continued engagement in aerobic exercise training sessions over time, previously developed neuroplasticity may become more robust. It has also been demonstrated that starting with exercise training before training for an intellectually challenging task can effectively prime the brain for new learning [41]. This enhanced neuroplasticity can subsequently lead to improved learning performance and cognitive function in the long term. This underscores the importance of longitudinal studies to comprehensively capture the extent of cognitive improvements over time.

The current study does not include the measurement of physical functions or fitness changes following a short-term intervention. However, it is well established that engaging in aerobic exercise over the long term yields both physical and cognitive benefits. If the findings, such as enhanced foreign language learning (FLL) outcomes and cognitive functions, remain consistent in a larger-scale study, the inclusion of aerobic exercise in advance of a foreign language learning program could potentially provide older adults with additional advantages in improving physical health and comprehensive cognitive skills simultaneously.

In addition to a larger-scale study and considering the long-term changes in physical function, future studies should explore the neurobiological mechanisms behind the combined exercise and language learning interventions. This includes incorporating additional measurements, such as fNIRS and EEG, to assess participants’ brain activity and changes in brain function within each group during the learning process. In the current study, only certain cognitive functions were examined under short-term interventions; the effects on other cognitive functions, such as executive functions and problem-solving, should also be studied.

## 5. Conclusions

In summary, we investigated how exercise influences foreign language acquisition by directly measuring learning outcomes and cognitive effects through a newly developed training paradigm. Our results suggest that a program like this can improve older adults’ Spanish learning performance. Additionally, language learning itself, with or without exercise, improves cognitive functions. However, significant improvements were observed only in the language learning-only (FLL) group, not in the combined aerobic exercise and language learning group (AE + FLL), which is inconsistent with previous studies that show combined exercise and cognitive training (CT) provide superior cognitive benefits compared to single training. This discrepancy could be due to our application of a language-learning method rather than traditional CT. Future studies need to explore the long-term effects of the combined strategy and the mechanisms behind this phenomenon, as well as the effects on other cognitive functions that are not included in this study.

## Figures and Tables

**Figure 1 brainsci-14-00572-f001:**
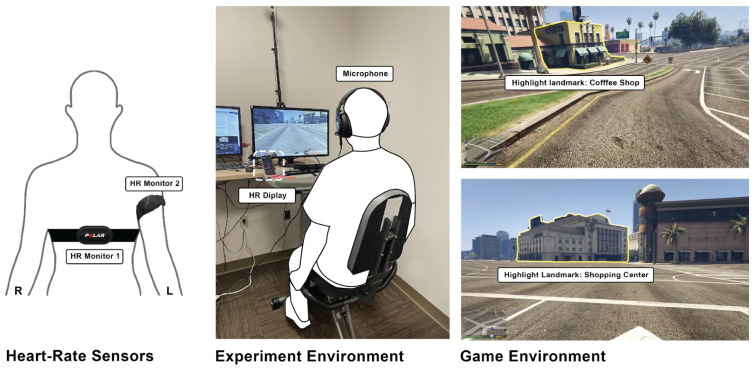
The experiment environment. The virtual scenes were originally from Grand Theft Auto V (GTA 5) [31] and were modified for the experiment.

**Figure 2 brainsci-14-00572-f002:**
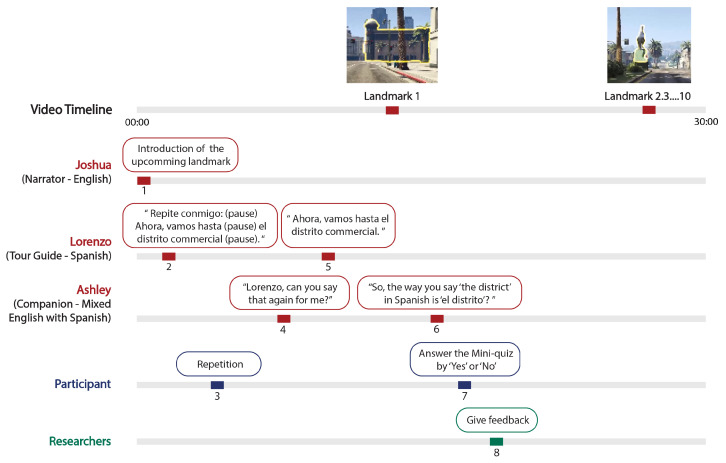
The Spanish learning process in the game involves interactions among game characters (NPCs) that provide language-learning cues to the participants, and the researchers offer feedback. The process follows the steps marked with numbers from one to eight. The translation of Lorenzo’s Spanish in English means "Now, let’s go to the commerical district".

**Figure 3 brainsci-14-00572-f003:**
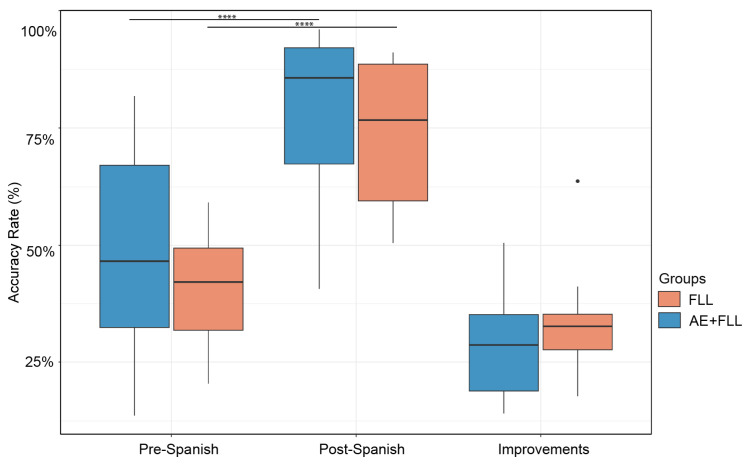
This figure shows participants’ Spanish learning performance in pretraining and post-training and the improvements in Spanish learning performance after the training sessions. The black dot indicates the outliers. The symbol ’****’ denotes significant differences (*p* ≤ 0.0001) between pre- and post-Spanish tests.

**Figure 4 brainsci-14-00572-f004:**
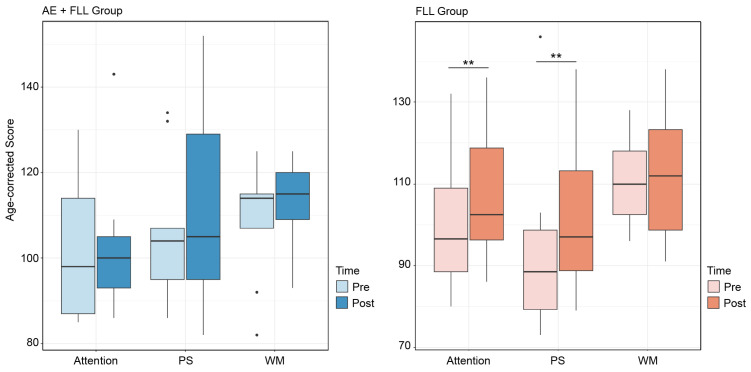
Pre- versus post-test cognitive scores for participants in the AE + FLL group and LL group. ‘PS’ stands for ‘Processing Speed’; ‘WM’ stands for ‘Working Memory.’ The black dot indicates the outliers. The symbol ’**’ denotes significant differences (*p* ≤ 0.01) between pre- and post-cognitive tests.

**Figure 5 brainsci-14-00572-f005:**
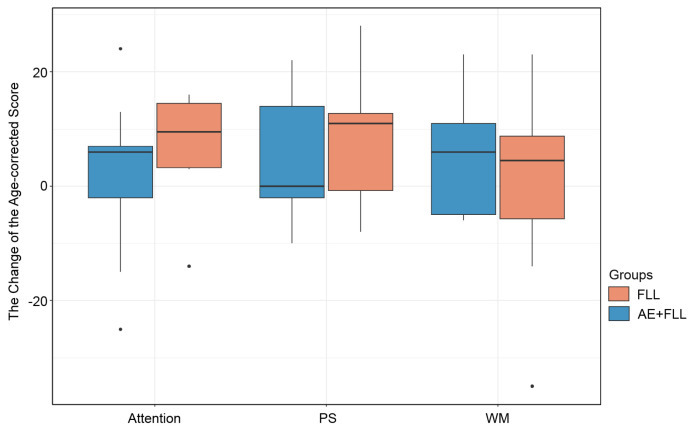
The change in cognitive performance after four training sessions. The black dot indicates the outliers.

**Figure 6 brainsci-14-00572-f006:**
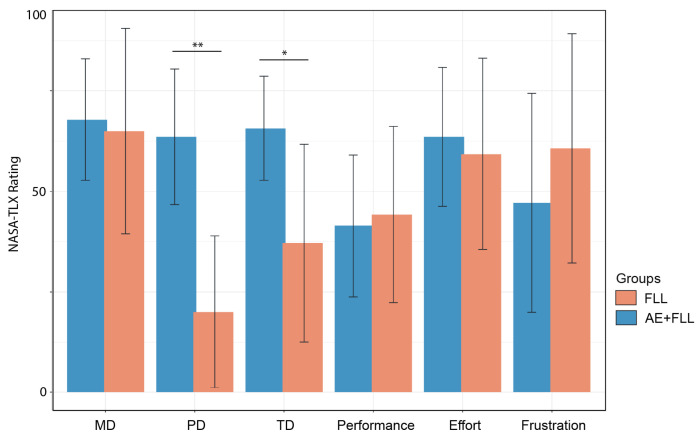
The perceived cognitive loads between the AE + FLL group and the FLL group, measured using the NASA-TLX scale. The symbol ’**’ denotes significant differences (*p* ≤ 0.01) between groups. The symbol ’*’ denotes significant differences (*p*≤ 0.05) between groups.

**Table 1 brainsci-14-00572-t001:** The variation in age, gender, MoCA scores, KBIT scores, existing Spanish proficiency, cognitive performance before intervention, average training session completion time, and the delay (in days) between the last training session and post-measurement for both AE + FLL and FLL groups.

Characteristics	AE + FLL (n = 10)	FLL (n = 10)	*p*-Value
Age	M ^1^ = 72.3 SD ^2^ = 3.3	M = 75.4 SD = 4.2	0.08
Gender	Female = 4 Male = 6	Female = 5 Male = 5	N/A ^3^
MoCA	M = 26.7 SD = 1.9	M = 26.6 SD = 1.4	0.897
KBIT	M = 115.2 SD = 13.5	M = 116.5 SD = 10.3	0.812
Pre-Spanish	M = 0.4 SD = 0.2	M = 0.4 SD = 0.1	0.401
Pre-Attention	M = 99.2 SD = 15.8	M = 99.8 SD = 16.3	0.934
Pre-PS	M = 104.5 SD = 16.7	M = 93.1 SD = 20.9	0.197
Pre-WM	M = 105.9 SD = 14.1	M = 111.2 SD = 11.1	0.364
Length (days)	M = 20.1 SD = 7.3	M = 18.8 SD = 7.1	0.692
Delay (days)	M = 3.3 SD = 2.1	M = 3.3 SD = 1.2	0.972

^1^ The mean is shown as ‘M’. ^2^ The standard deviation is shown as ‘SD’. ^3^
*p*-value is not calculated for gender because of the small sample size.

## Data Availability

Data are unavailable due to privacy.

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
