# Peer review of "The Influence of Separate and Combined Exercise and Foreign Language Acquisition on Learning and Cognition"

_brainsci, 2024, doi:10.3390/brainsci14060572_

Round 1

Reviewer 1 Report

Comments and Suggestions for Authors

The authors investigated the combined effect of aerobic exercise (AE) and foreign language learning (FLL) in older adults. The FLL was conducted using a virtual world tourism scenario. The results showed that the FLL improved the knowledge of foreign language, but there was no additional improvement from aerobic exercise. The FLL also improved attention and processing speed, but no improvement was observed in the combined intervention. This study helps to provide a novel intervention to improve brain health and cognitive function. While I am interested in this study, I have some concerns. I would like the authors to address the following points.

Regarding the experimental protocol, the authors set four days of learning session. As referred to the Length in Table 1, it seems that the intervention interval was quite spaced. In addition, although there is no statistical difference between the groups, the S.D. of the Length is so large. Does this imply that the interval between each intervention varied widely across participants? Is it possible that this variation in interval is causing variation in the learning effect? Is this an appropriate protocol for a learning experiment? I would like comments on the validity and justification of the learning protocol.

Did the participants check and control their heart rate themselves during the exercise? If so, I think that just doing exercise and FLL at the same time has a higher cognitive load and that is further increased by the need to pay attention to and control their heart rate. I wonder if this additional cognitive load for controlling heart rate masks exercise effect.

Why did the authors conduct t-test repeatedly, instead of an ANOVA?

Line 321: I am concerned that the sentence “working memory benefits more from the combined strategy” is an overstatement. While I understand that there is a trend of improvement with AE+FLL, but there is no difference and no trend in the group comparison, it cannot be said that the combined strategy is more beneficial.  

Reviewer 2 Report

Comments and Suggestions for Authors

The article ‘The Influence of Separate and Combined Exercise and Foreign Language Acquisition on Learning and Cognition’ aims to investigate how exercise influences foreign language acquisition by directly measuring learning outcomes and cognitive effects. The authors should add the purpose of the article and hypotheses at the end of the Introduction. The purpose of the article is stated by the authors only in Discussion.

The topic that is developed in this article is very important and relevant from both fundamental and applied points of view. In the Introduction, the authors review the relevant literature and highlight the existing gaps.

Compared with other published material, this study adds to the subject area the information about the development of paradigms integrating language learning with exercise limits research on combined effects in older adults.

The methodology of the study is fine.

The conclusions in Discussion are consistent with the evidence and arguments presented. But the authors should add a separate section with conclusions.

The references are appropriate.

Reviewer 3 Report

Comments and Suggestions for Authors

Thank you for the opportunity to review this interesting original articles. It is about the study of a dual-task training in which an exercise is combined to the learning of a language. The manuscript is overall interesting, well written and supported with tables and figures. The topic is actual and of interest for the scientific community and not. My opinion on the manuscript is positive, but I have some comment before I can accept it for publication.

Line 32: “The concurrent or simultaneous performance of physical exercise and cognitively challenging activities is known as combined, multidomain, or dual-task training”. Please, support the sentence with references and implement the topic with a deeper explanation of the dual task concept. Interesting works on the work are: 

analysis”. Journal of Aging and Physical Activity

-Effects of dual tasks and dual task training on postural stability: A systematic review and meta analysis”. Clinical Interventions in Aging.

Line 45: “Still, limited investigations explored the combination of AE and FLL in older adults.”. Please, support the sentence with reference

Line 48 “Previous studies have examined the preferences and challenges faced by older adults during exercise, travel, and studying foreign languages. Researchers”. The Authors wrote “previous studies”. Please, specify the studies.

Line 54-61: this part could be included in the methods section, not in the introduction.

Line 67-80: this part could be included in the methods section, not in the introduction.

Methods:

Please, add if the study followed the declaration of Helsinki. Please, specify if the participants signed the consent form for data treatment before the data collection.

Most of the methods section is not supported by the literature. It should be interesting to support the test and the procedure with references that show the validity and feasibility.

Table 1. Please, provide below the table the abbreviation adopted in its long form associated with the abbreviation. Example: female: f; male: m; standard deviation: SD…

Please, move table 1 in the results section, immediately after table 1 appear in the text.

Results: well-written and presented.

Discussion: please add the possible future studies on this topic. Please, add the paragraph “conclusion”.

Round 2

Reviewer 1 Report

Comments and Suggestions for Authors

Thank you for your comments and updating the manuscript. There are no further revision requests on the manuscript.

Reviewer 3 Report

Comments and Suggestions for Authors

Thank you for addressing my comments. The manuscript is sufficiently improved